# Study of New Therapeutic Strategies to Combat Breast Cancer Using Drug Combinations

**DOI:** 10.3390/biom8040175

**Published:** 2018-12-14

**Authors:** Ana Correia, Dany Silva, Alexandra Correia, Manuel Vilanova, Fátima Gärtner, Nuno Vale

**Affiliations:** 1Laboratory of Pharmacology, Department of Drug Sciences, Faculty of Pharmacy, University of Porto, Rua de Jorge Viterbo Ferreira 228, 4050-313 Porto, Portugal; up201607999@fc.up.pt (A.C.); danys@ua.pt (D.S.); 2Department of Molecular Pathology and Immunology, Institute of Biomedical Sciences Abel Salazar (ICBAS), University of Porto, Rua de Jorge Viterbo Ferreira 228, 4050-313 Porto, Portugal; fgartner@ipatimup.pt; 3i3S, Instituto de Investigação e Inovação em Saúde, University of Porto, Rua Alfredo Allen 208, 4200-135 Porto, Portugal; alexandra.correia@ibmc.up.pt (A.C.); vilanova@icbas.up.pt (M.V.); 4Department of Imuno-Physiology and Pharmacology, Institute of Biomedical Sciences Abel Salazar (ICBAS), University of Porto, Rua de Jorge Viterbo Ferreira 228, 4050-313 Porto, Portugal; 5Institute of Molecular and Cell Biology (IBMC) of the University of Porto, Rua Alfredo Allen 208, 4200-135 Porto, Portugal; 6Institute of Molecular Pathology and Immunology of the University of Porto (IPATIMUP), Rua Júlio Amaral de Carvalho 45, 4200-135 Porto, Portugal

**Keywords:** breast cancer, drug repurposing, drug combinations, 5-fluorouracil, cell viability, epithelial-mesenchymal transition

## Abstract

Cancer is a disease that affects and kills millions of people worldwide. Breast cancer, especially, has a high incidence and mortality, and is challenging to treat. Due to its high impact on the health sector, oncological therapy is the subject of an intense and very expensive research. To improve this therapy and reduce its costs, strategies such as drug repurposing and drug combinations have been extensively studied. Drug repurposing means giving new usefulness to drugs which are approved for the therapy of various diseases, but, in this case, are not approved for cancer therapy. On the other hand, the purpose of combining drugs is that the response that is obtained is more advantageous than the response obtained by the single drugs. Using drugs with potential to be repurposed, combined with 5-fluorouracil, the aim of this project was to investigate whether this combination led to therapeutic benefits, comparing with the isolated drugs. We started with a screening of the most promising drugs, with verapamil and itraconazole being chosen. Several cellular viability studies, cell death and proliferation studies, mainly in MCF-7 cells (Michigan Cancer Foundation-7, human breast adenocarcinoma cells) were performed. Studies were also carried out to understand the effect of the drugs at the level of possible therapeutic resistance, evaluating the epithelial-mesenchymal transition. Combining all the results, the conclusion is that the combination of verapamil and itraconazole with 5-fluorouracil had benefits, mainly by decreasing cell viability and proliferation. Furthermore, the combination of itraconazole and 5-fluorouracil seemed to be the most effective, being an interesting focus in future studies.

## 1. Introduction

Cancer is the second-leading cause of death behind heart disease [1], accounting for an estimation of 9.6 million deaths in 2018 [2].

When dealing with this disease, there are three important priorities, which are, in order of importance: prevention, early detection and total eradication. The treatment of cancer, particularly, is a very complex issue, with three usual modes: surgery, radiotherapy and pharmacotherapy [3].

Concerning pharmacotherapy, the ideal drug is one that selectively kills the neoplastic cells, minimizing adverse effects. However, the differences between normal and neoplastic cells reside, usually, in a quantitative way, such as greater or lesser activation of signaling pathways. Therefore, pharmacotherapy of cancer is difficult to achieve in a totally successful way [4]. Nevertheless, survival of oncological patients has improved significantly in the last years, mainly because of multidisciplinary care, improved chemotherapeutic agents, introduction of targeted therapy, and the incorporation of palliative care services [5]. However, despite the advances mentioned above, many patients still fail therapy, explained by the presence of an intratumoral heterogeneity and several drug resistance mechanisms, such as deregulation of apoptosis, activation of prosurvival signaling and epithelial-mesenchymal transition (EMT) [6]. Intense pharmacological research is being made about oncological therapy, originating an explosion of costs. Despite this big investment, there is little output for a huge pharma research and development spending. Thus, there is a requirement for more effective cancer drugs. To face this problem, an interesting approach named drug repurposing is being increasingly applied [7]. Highlighting the importance of this approach, the global market for drug repurposing reached €20.7 billion in 2015 and is projected to reach €26.6 billion by 2020 [8]. Additionally, another important response to the problems in cancer therapy encompasses the use of drug combination therapies [9].

Drug repurposing is a methodology to identify a new indication for already existent drugs. It allows lower costs and a shorter time until approval of the drug, than developing a drug de novo, because information regarding side effects, pharmacokinetics and interaction with other drugs has been collected [10]. Cytostatic and/or cytotoxic activity within a wide range of drug classes other than cancer has been demonstrated in several studies. Therefore, the study of the arsenal of drugs approved for non-cancer indications might offer effective treatment options for cancer patients. Commonly, ideal drugs for repurposing share several characteristics: they should be well-known drugs, often available as generics, the toxicology profile of the drug must be good, there should have a plausible mechanism of action, relevant for the condition in question, and evidence of efficacy at physiological dosing [9]. Many currently used drugs have, at least, some actions that may be useful in cancer treatment. Particularly, a lot of off-patent drugs have shown some evidence of anticancer effects, in which about 50% are supported by relevant human data and 16% are supported by data from at least one positive clinical trial [11]. Briefly, for oncologic purposes, drugs like aspirin, itraconazole, verapamil, chloroquine and all-trans retinoic acid have shown anticancer activity in, at least, one randomized clinical trial [12,13].

Concerning drug combination approach, it is known that a disease is interpreted as a set of molecular pathways that interconnect, having a bigger susceptibility to the simultaneous action of several drugs. This makes possible to study drug combinations in greater depth [14]. Combining drugs has several advantages: decreased toxicity, better efficacy, decreased dosage at an equal or increased level of efficacy, and counter drug resistance [15]. Due to these advantages, drug combinations represent an interesting and increasingly used approach that has become a standard for the treatment of a wide range of diseases, such as cancer and infectious diseases [14]. Particularly in cancer, numerous clinical trials testing combinations that include chemotherapy drugs, radiation therapy, hormonal therapies, molecularly targeted therapies, and immunotherapies are being carried out [16], with a crescent focus on the combination of cytotoxic chemicals and biotherapies [17].

Combining repurposed pharmaceutical agents with other chemotherapeutic agents has also shown promising results, useful when traditional anticancer monotherapy has failed to provide a safe and tolerable treatment for cancer patients [18]. An example of these kinds of combinations is nitroglycerin in combination with the chemotherapeutics vinorelbine and cisplatin. In this case, one randomized Phase II trial demonstrated improved overall survival of patients with non-squamous cell lung cancer [18].

Breast cancer is a collection of breast diseases that have distinct histopathologies, genetic, genomic variations, and clinical outcomes [19], being the second most common cancer worldwide, the most frequent cancer in women and the fifth cause of death from cancer overall. In the past two decades, the rates of breast cancer mortality have declined by approximately 30%, with corresponding improvements in five-year overall survival rates to 90%. However, despite these advances, metastatic breast cancer remains a challenge to treat, with an estimated five-year overall survival rate of only 23% [20].

The choices of treatment regimens for breast cancer is a very complex issue, being difficult to have a universally accepted treatment, since each case is a case and the choice of the best therapy should be tailored to each person. Thus, a lot of expertise is necessary to evaluate the best therapy for each case [21].

For early-stage breast cancer, surgery is considered the gold-standard treatment [22]. On the other side, the adjuvant treatment’s main goal is to treat metastatic disease. This treatment consists of radiation therapy and systemic therapy (including a variety of chemotherapeutic, hormonal and biologic agents) [23]. There are a lot of approved drugs for breast cancer and, in addition, there are a great variety of drugs that have the potential to be repurposed for its treatment [24]. However, to date, no repurposed drugs have been approved for the treatment of this kind of cancer. An example of a potential repurposed drug is the beta-blocker propranolol. A study showed that this beta blockade reduced tumor proliferation by 66%, in early stage breast cancer, by accessing Ki67. Chloroquine is another example, with very promising results in breast cell lines and Phase I trials, either alone or in combination with other drugs [25].

Despite a lot of approved drugs and potential drugs to repurpose, breast cancer remains a major healthcare issue. Highly qualified healthcare providers are necessary for proper surgical treatment. Additionally, advanced treatment approaches that involve radiation are difficult to achieve in developing countries, particularly. Adequate systemic treatments, management of potential severe effects and new targeted therapies are costly, and the most sophisticated treatments require advanced and costly pathology. In addition, the problem of drug resistance, transversal to most of the cancer types, remains an important issue of study and combat [26].

Thus, this project aimed to investigate the effect of drugs with potential to be repurposed for breast cancer therapy, in combination with an already used drug in this type of therapy (5-fluorouracil), so that the combination of these drugs presented benefits, when compared to the single drugs of the combination.

5-Fluorouracil (5-FU) was chosen as the reference drug of this study, mainly because it is a potential drug to be used in combination regimens in breast cancer therapeutics, with the major aim of improving its efficacy, as well as its known toxicological profile. It is a heterocyclic aromatic organic compound with a structure similar to that of the pyrimidine molecules of DNA and RNA. This drug and their metabolites are responsible for RNA and DNA damage, acting on S phase of cell cycle [27].

Studies addressing combinations of drugs using drugs with non-cancer indications have shown positive results in relation to cancer therapy, not exclusively for breast cancer, but also for other kinds of cancer [28]. By integrating all the obtained results, it was notorious that the chosen drug combinations showed a tendency to be more effective than the most effective drug of each combination. Possibly, these drug combinations may act primarily in the arrest of cell proliferation. Also, it is important to highlight that the collective of all results pointed out that the combination of 5-FU and itraconazole was the most promising combination.

## 2. Materials and Methods

### 2.1. Drug Solutions

For the treatment of cells with the several drugs under study (5-FU, verapamil, itraconazole, isoniazid, tacrine, aspirin, cimetidine, chloroquine, losartan, all from Sigma-Aldrich (Sigma-Aldrich Quimica, S.L., Sintra, Portugal) and pravastatin, from Cayman Chemical Company, Ann Arbor, MI, USA), all the compounds were dissolved in autoclaved water, except itraconazole, that was dissolved in dimethyl sulfoxide (DMSO), since it did not present solubility in water. A stock solution of each compound was prepared at a concentration of 10 mM and, except for itraconazole, two stock solutions with a concentration of 25 mM and 50 mM were prepared, because this drug was dissolved in DMSO, which has significant toxicity to cells after a percentage of 0.2%, a percentage that was never exceeded in this work. All these stock solutions were conserved on the freezer at −26 °C. Depending on the purpose of the assay, test compounds were used in concentrations that range from 1 μM to 100 μM, dissolved in culture medium right before contact with cells. The respective concentrations used in each assay are presented in the Results section. The test compounds applied to the cells vary with the purpose of the experiment, also specified in the above-mentioned section.

### 2.2. Cell Culture

The experimental work was carried out mainly in the MCF-7 cell line (ATCC, American Type Culture Collection, Manassas, VA, USA). Additional experiments were carried out in MCF-10A (ATCC—American Type Culture Collection). MCF-7 and MCF-10A cells were incubated at 37 °C in a humidified atmosphere with 95% of air and 5% of CO2. MCF-7 cells were cultivated in Dulbecco’s Modified Eagle Medium (DMEM), supplemented with 10% fetal bovine serum (FBS) and 1% of a mixture of penicillin/streptomycin (1000 U/mL; 10 mg/mL). MCF-10A were cultivated in DMEM/F-12, supplemented with the same supplements described above plus 2 μg/mL of human insulin (NovoNordisk, Bagsværd, Denmark), 20 ng/mL of epidermal growth factor (EGF) and 1 μM of hydrocortisone. For maintenance, the cells were cultured in a monolayer, being subcultured 2–3 days per week, and 1 day per week in the case of MCF-10A, the last cell line growing slowly. All the experiments were carried out with cells to 70–80% confluence.

### 2.3. MTT Reduction Assay

MCF-7 and MCF-10A cells were plated in 96-well plates with a seeding density of 3.0 × 10^4^ cells/mL, maintained at an incubator at 37 °C, for 24 h. After this time, different treatments were added to the cells at different concentrations, for 48 h or 72 h, depending on the purpose of the experiment. The cells were maintained at 37 °C during the referred time and, after that, the cell medium was removed and 100 μL of (4,5-dimethylthiazol-2-yl)-2,5-diphenyltetrazolium bromide (MTT) solution (0.5 mg/mL in PBS) was added to each well. Then, the cells were incubated at 37 °C for 3 h in a light-protected manner. At the end of this time, MTT solution was removed and 100 μL/well of DMSO was added, with the purpose of solubilization of formazan crystals. The last step consisted of absorbance readings at 570 nm in an automated microplate reader (Sinergy HT, BioTek Instruments, Winooski, VT, USA).

### 2.4. Immunocytochemistry

MCF-7 cells were seeded in T25 cm^2^ flasks with a density of 4.7 × 10^4^ cells/mL, converted from the density used in 96-well plates (3.0 × 10^4^ cells/mL). After that, cells were incubated for 24 h at 37 °C. Then, the different test compounds were added and acted for 48 h. After this time, the cells were included in cell blocks (one for each different treatment) and different slides were made from each block. Then, deparaffinization by submerging the slides twice in xylene was performed, for 5 min each time, followed by hydration in alcohol at decreasing concentrations (100%, 95% and 70% alcohol) until rinsing in water. After that, antigen retrieval (unmasking) was performed by using a Retrieval Solution (10% in water), 20 min in a water-bath at 100 °C. The next step consisted in endogenous peroxidase block, by incubation of the slides in a Peroxidase Block Solution for 5 min. Then, the slides were washed twice in TBS for 5 min. Incubation with protein block for 5 min followed, and the slides were washed in TBS 2× for 5 min. Afterwards, the slides were incubated with the mouse antibodies anti-cytokeratin (pan) (1:1200 in BSA 5%), anti-E-cadherin (1:50 in BSA 5%) and anti-vimentin (1:500 in BSA 5%), overnight, at 4 °C. The slides were washed twice in TBS, for 5 min and the post primary was added, following an incubation time of 30 min. Once again, the slides were washed twice in TBS, for 5 min. After that, they were incubated for 30 min with the Polymer, washed twice in TBS for 5 min and, to each slide, 150 μL of a solution of 50 μL of DAB Chromogen to 1 mL of DAB Substrate Buffer was added to each slide. Finally, the slides were rinsed in water, counterstained in hematoxylin during 1 min, washed again for 5 to 10 min, dehydrated (at increasing concentrations of alcohol, 70%, 95% and 2× 100%), diaphanized (2× in xylene), and the sections were mounted. The slides were observed on a Nikon Eclipse E600 microscope, coupled to a digital camera (Nikon Digital Sight DS-Fi2, Tokyo, Japan). The images were treated with Imaging Software NIS-Elements AR Version 4.30.0 (Nikon, Tokyo, Japan).

### 2.5. Annexin V-FITC and Propidium Iodide Staining

MCF-7 cells were seeded in 96-well plates with a seeding density of 1.0 × 10^6^ cells/mL, incubated at 37 °C for 24 h. After this time, the different treatments were added to the cells that were incubated for 3 or 8 h. Then, cells were trypsinized, washed with Hank’s balanced salt solution (HBSS, 2%FBS), centrifuged for 5 min, 400× *g*, and stained with the eBioscience Annexin V Apoptosis Detection Kit Fluorescein Isothiocyanate (FITC) (Waltham, MA, USA), as follows: Annexin V FITC (1 μL per well) in Binding Buffer (50 μL per well), 15 min at room temperature, in the dark. Then, cells were transferred to a test tube and 2 μL of propidium iodide (PI) were added to each sample, 5 min before analysis in a flow cytometer (Beckman Coulter Epics XL and BD FACSCanto II; Brea, CA, USA and, San Jose, CA, USA, respectively). The acquired data was analyzed using FlowJo (V10) analysis software (Ashland, OR, USA).

### 2.6. CFSE Labeling of Cells

MCF-7 cells were resuspended in PBS with 0.1% BSA at a density of 2.0 × 10^6^ cells/mL. The CellTraceTM carboxyfluorescein succinimydil ester (CFSE) Cell Proliferation Kit (Molecular Probes, Invitrogen, Carlsbad, CA, USA) was used for cell labelling. A CFSE stock solution (5 mM in DMSO) stored at −20 °C was thawed and diluted in PBS with 0.1% BSA to a final concentration of 10 μM. Next, 2.0 × 10^6^ cells/mL were labelled by the addition of the same volume of the prepared CFSE staining solution and incubated at 37 °C for 10 min. The staining solution was neutralized with three volumes of cold complete medium (DMEM, 10% FBS) and incubated for 5 min on ice. Then, cells were centrifuged at 400× *g* for 5 min, the supernatant was removed, and the cells were washed one more time with complete RPMI medium. The cell pellet was resuspended in complete medium at a density of 1.0 × 10^6^ cells/mL and cells were seeded in 96-well plates for 3 h. After that, the medium was aspired and test compounds, dissolved in the culture medium, were added to cells, that were incubated at 37 °C for approximately 72 h. The final step consisted of washing and resuspension of cultured cells in HBSS (2% FBS). Five min before reading, 2 μL of PI were added to each cytometer tube (that represents each condition) for dead cell exclusion. Finally, cell proliferation was determined by flow cytometry (Beckman Coulter Epics XL, Brea, CA, USA) and the data was analyzed using FlowJo (V10) analysis software.

### 2.7. Statistical Analysis

Statistical analysis was performed in all experiments, only in the case of a number of independent experiments equal or bigger than 3 (*n* ≥ 3). The results are expressed as arithmetic mean ± standard error of the mean (SEM), except in one case, where results are expressed as arithmetic mean ± standard deviation (SD), explicit in the subtitles of the graphs. Differences between treated cells and corresponding untreated control were tested using one-way ANOVA followed by Dunnett’s test. Differences between the drug combination and the respective individual drug of that combination that produces more advantageous effects in terms of cell viability reduction were tested by Student’s *t*-test. Differences were considered to be significant when *p* value < 0.05. One-way ANOVA followed by Dunnett’s test and Student’s *t*-test were performed by using SigmaPlot 12.0 (San Jose, CA, USA) and GraphPad Prism 7 (San Diego, CA, USA), respectively. It is important to note that, for all experiments, no differences were observed between control with/without DMSO.

## 3. Results

### 3.1. Drug Screening in MCF-7 Cells

Based on several studies about potential drugs to repurpose [10,12,29,30] and interests of the investigational group, nine drugs, each in combination with the reference drug (5-FU), were tested in MCF-7 cell line, in order to make an initial screening of potential drugs to be used in combination with 5-FU. Each drug was used in a concentration of 50 μM, thus, in a ratio of 1:1 when combined with 5-FU, being in contact with cells for 72 h. The results were obtained by MTT methodology (Figure 1).

In this particular screening assay, the criterion for the choice of drug combinations for the continuity of the project was that the combination of drugs was more advantageous in terms of reduction of cell viability than the two drugs in the combination, where the potentially repurposed drug was more efficient than 5-FU. The combination was more effetive than drugs separated.

Analyzing the obtained results, it was possible to observe that chloroquine was more effective in terms of cell viability reduction than all the other drugs and drug combinations (6.5 ± 0.4% of cellular viability). Thus, as the aim of this work was to study a beneficial drug combination in comparison with individual drugs of the combination, chloroquine was excluded from the next steps. Importantly, the combinations of 5-FU with aspirin, losartan, cimetidine, pravastatin, isoniazid and tacrine did not show an advantage in terms of reduction of cell viability, relative to both single drugs of the combination, being also excluded from this study. However, two drug combinations were advantageous: 5-FU combined with verapamil and itraconazole, chosen for the continuity of this project. The exposure of MCF-7 cells to 5-FU combined with verapamil and itraconazole, for 72 h of contact with cells, resulted in a cell viability reduction (in comparison with the drug with more effect on viability reduction of that combination, the potential repurposed drug) of 23% and 17%, respectively. With 5-FU + verapamil, cell viability was 12.1 ± 4.4%, whereas with 5-FU + itraconazole was 24.5 ± 5.2%. In both cases, the differences were considered statistically significant.

### 3.2. Comparison of Cellular Viability between MCF-7 and MCF-10A Cell Lines

To compare the effects of the chosen drug combinations in a tumoral cell line (MCF-7) and a non-tumoral cell line (MCF-10A), both cell lines were exposed to 50 μM of each drug, for 72 h. The results, for each cell line, were obtained by MTT methodology (Figure 2).

It was observed that, in general, the drugs had no or very little effects on the viability of MCF-10A cells, contrasting with the effects on viability of MCF-7 cells. Taking into account the two combinations, 5-FU combined with itraconazole led to the lowest values of cell viability in MCF-10A, 63.6 ± 8.5%, whereas 5-FU combined with verapamil led to similar values of cellular viability: 86.3 ± 3.2%. All the single drugs and combinations led to effects on cell viability reduction of MCF-7 cells much more pronounced, compared with MCF-10A cell line.

### 3.3. Concentration-Effect Curves and IC_50_ Determination of the Drugs in MCF-7 Cells

In order to determine the IC_50_ (half maximal inhibitory concentration) values of 5-FU, verapamil and itraconazole on the viability of MCF-7 cells, cells were exposed to increasing concentrations of these drugs (1, 3, 5, 10, 25, 50 and 100 μM), converted in logarithm of concentrations. The drugs were in contact with cells for a time of 48 h, and cellular viability was determined by MTT reduction assay. The obtained values allowed the creation of concentration-effect curves (Figure 3, Figure 4 and Figure 5) to calculate IC_50_ values (Table 1).

The IC_50_ of the drugs was calculated by using the normalized data (between 0% and 100%). Thus, it was defined as the concentration of the drug that inhibits 50% of cellular viability. However, this method of calculation of IC_50_ has limitations, as reported in the Discussion Section. This value was calculated using GraphPad Prism 7 software and the option “Analyze Data, nonlinear regression (Curve fit)”.

### 3.4. Effect of 5-FU Fixed in Its IC_50_ Value and Variation of the Repurposed Drugs Concentration in Values around Their IC_50_ Values, on Viability of MCF-7 Cells

The IC_50_ value obtained for 5-FU (approximately 11.8 μM) was fixed, and the concentration of the other two drugs varied by values around the obtained IC_50_ value for each drug (approximately 29 and 6 μM for verapamil and itraconazole, respectively). It is important to note that the IC_50_ value obtained for itraconazole was initially 6 μM and this was the value of IC_50_ considered to this drug. However, in a final phase of the work, the experiment of generating a concentration-effect curve for itraconazole was repeated and a new value of IC_50_ (2.08 μM) was obtained.

The drugs were in contact with cells for a time of 48 h and cellular viability was determined by MTT reduction assay (Figure 6).

The lowest values of cellular viability were obtained for concentrations of 55 and 3 μM, for verapamil and itraconazole, (10.4 ± 0.6%; 34.9 ± 2.7%) respectively. A tendency of lower values of cellular viability with an increase of verapamil concentration was observed, whereas with itraconazole, it seemed that above 3 μM, the concentration of the drug had little or no effect on reduction of the cellular viability. Clearly, verapamil appeared to be the drug which was more affected by differences in concentration.

### 3.5. Effect of Concentrations of 11.8, 55 and 3 μM of 5-FU, Verapamil and Itraconazole, Respectively, on the Viability of MCF-7 Cells, for 48 and 72 h

Further, we tested the effect of 5-FU, verapamil, itraconazole and the respective combinations of the last two drugs with 5-FU on the viability of MCF-7 cells, using 5-FU in a concentration of 11.8 μM (IC_50_ value), and verapamil and itraconazole in a concentration of 55 and 3 μM, respectively, acting on cells for a time of 48 h or 72 h (Figure 7). The concentrations used for the repurposed drugs were selected taking into account the experiment described in Section 3.4, being the concentrations that gave lower values of cellular viability in combination with 5-FU fixed on its IC_50_ value.

Analyzing the obtained results, for 48 h-treatment with the drugs, it was concluded that the drug combinations seemed to be only slightly advantageous, compared with the single drug with more effect on the reduction of cell viability of each combination. In the case of verapamil combined with 5-FU, the single drug with more effect on cellular viability reduction was 5-FU, with values of cell viability of 71.0 ± 2.1% (combination) and 76.9 ± 2.6% (5-FU). In the case of itraconazole combined with 5-FU, values of 35.8 ± 1.1% of cellular viability contrasted with values of 41.4 ± 2.2% of itraconazole, the single drug of the combination with more effect on cellular viability reduction. Thus, even though there were no notable differences, there was a tendency of the combination to be more effective than the single drug with more effect on the reduction of cell viability of that combination.

For 72 h, it was possible to observe that the combinations of 5-FU + verapamil (49.7 ± 2.5% of cellular viability) and 5-FU + itraconazole (32.5 ± 5.5% of cellular viability), compared with the single drug of each combination with more effect on the reduction of cell viability, with 7% and 11%, respectively, keeping the same tendency above mentioned. It was also noted that when comparing different time points (48 vs. 72 h), the drugs and combinations reduced cell viability more markedly at 72 h.

At both 48 and 72 h, the lowest values of cellular viability were obtained with 5-FU + itraconazole. The increase of independent experiments offered a strong probability of observation of statistically significant differences, since there is a clear tendency of advantageous effects of drug combinations (in both 48 and 72 h), relative to individual drugs.

### 3.6. Effect of 50 μM of Each Drug on the Viability of MCF-7 Cells, for 48 and 72 h

The effect of 5-FU, verapamil and itraconazole and the respective combinations of the last two drugs with 5-FU on the viability of MCF-7 was tested, using all the drugs in a concentration of 50 μM, the same concentration used in the initial screening assay. The drugs were in contact with cells for 48 h and 72 h (Figure 8).

For 48 h of actuation of the drugs and combinations, it was observed that the two combinations, compared with the single drug, with more effect on the reduction of cell viability with 3% (in the case of verapamil combined with 5-FU) and 6% (in the case of itraconazole combined with 5-FU). For 72 h, this reduction was 23% and 17% respectively.

At 48 h of drug actuation, the lowest values of cellular viability were obtained with 5-FU + itraconazole (47.9 ± 6.5% of cell viability). On the other hand, at 72h, not only higher differences between the most effective single drug of the combination and the respective combination, but also the lowest values of cellular viability were achieved with the 5-FU + verapamil (12.1 ± 2.2% of cell viability). It was also noticeable that, when comparing different time points (48 vs. 72 h), all the drug combinations and single drugs reduced cell viability more markedly at 72 h than at 48 h.

### 3.7. Comparison of Different Concentrations of the Drugs on the Viability of MCF-7 Cells, at 48 and 72 h

The results presented below represent the results obtained in Section 3.5 and Section 3.6, with the purpose of comparing the different concentrations tested in different time points (48h and 72 h) (Figure 9).

Analyzing the results obtained for 48 h-treatment, it was observed that the different concentrations did not produce very different effects in the case of 5-FU and 5-FU combined with verapamil. On the other side, in the case of verapamil, itraconazole and 5-FU combined with itraconazole, with concentrations of 50 μM, it was possible to observe that cell viability was less reduced.

For 72 h-treatment, in general, there was marked reduction in cellular viability, compared with 48 h. Once again, it was observed that the different concentrations did not produce very different effects, except in the case of verapamil and 5-FU combined with verapamil, in which concentrations of 50 μM led to an increase in cell viability reduction.

Taking into account all the different concentrations and time points, the drug combination that appeared to be more effective in terms of cell viability reduction was 5-FU + verapamil in a concentration of 50 μM of each drug, at 72 h (12.1 ± 2.2%). However, in all other conditions, 5-FU + itraconazole seemed to be the more effective drug combination, leading to reduced cellular viability values.

### 3.8. Evaluation of the Transition from an Epithelial to a Mesenchymal Phenotype, in MCF-7 Cells

Immunocytochemistry was used to assess the epithelial markers cytokeratins AE1/AE3 and E-cadherin, and the mesenchymal marker, vimentin, aiming at evaluating whether the resistant cells evolved from an epithelial to a mesenchymal state. MCF-7 cells were treated with 5-FU, verapamil, itraconazole, and the respective combinations of the last two drugs with 5-FU, acting on cells for a 48 h-period. 5-FU was used in a concentration of 11.8 μM and verapamil and itraconazole were used in a concentration of 55 and 3 μM, respectively. The obtained images are presented below (Figure 10).

It was possible to observe that, in all conditions, the cells were positively stained for the epithelial markers, AE1/AE3 and E-cadherin, and negatively marked for the mesenchymal marker, vimentin. Other findings were the presence of fewer cells, damaged cells, cells with vacuoles and dead cells. These findings will be discussed in the next section.

### 3.9. Analysis of Cell Death in MCF-7 Cells, by Annexin V/PI Staining

In order to understand the mechanism by which drug combinations act, we further tested them on the death of MCF-7 cells, using Annexin V/PI staining evaluated using flow cytometry. 5-Fluorouracil was used in a concentration of 11.8 μM (IC_50_ value) and verapamil and itraconazole were used in a concentration of 55 and 3 μM (with lower value from Figure 6), respectively, acting on cells for 3 h or 8 h. The gating strategies (Figure 11), as well as the results (Figure 12 and Figure 13) are presented below. Results are expressed as a percentage of living cells, apoptotic cells and cells in late apoptosis/necrosis, compared to untreated control.

The analysis of the obtained results, however preliminary, allowed concluding that, at 3 h drug exposure, with all the single drugs and combinations, the percentages of living cells were not very different from the control (100%). However, concerning the percentage of apoptotic cells, the highest value was obtained with 5-FU + itraconazole (126.8 ± 23.9%), while with all the other drugs and drug combinations, the values were not very different from control values. More pronounced differences were obtained concerning late apoptosis/necrosis: verapamil (139.2 ± 10.2%) and 5-FU + verapamil (130.2 ± 20.7%) led to the highest values, compared to control values.

At 8 h of treatment, the percentage of living cells also remained at values near to control values with all the treatments. Apoptosis was verified in more extension with the treatment with 5-FU (159.4 ± 3.5%), verapamil (192.9 ± 55.1%), 5-FU + verapamil (190.4 ± 13.9%) and itraconazole (161.4 ± 8.7%), while with other conditions, values similar to that of control were obtained. Concerning late apoptosis/necrosis, the highest values were obtained with verapamil (149.6 ± 6.16%) and 5-FU + verapamil (130.7 ± 2.3), whereas the values obtained with the other conditions were similar to the control value. In general, 8 h upon incubation, there was an increase of apoptosis, late apoptosis and necrosis when compared with 3 h incubation samples. Specifically, regarding the drug combinations, it is relevant to highlight the increase of approximately 102% in apoptosis with 5-FU + verapamil at 8 h (vs. 3h) and the decrease of approximately 20% in apoptosis, with 5-FU + itraconazole.

### 3.10. Analysis of Proliferation in MCF-7 Cells Using CFSE Staining

The drug combinations were further tested for their effect on the proliferation of MCF-7 cells, using flow cytometry and CFSE staining. 5-Fluorouracil was used in a concentration of 11.8 μM (IC_50_ value) and verapamil and itraconazole were used in a concentration of 55 and 3 μM, respectively, acting on cells for a time of 72 h, in order to allow enough time for cells to proliferate. The results (Figure 14 and Figure 15), as well as the performed gating strategies (Figure 16), are presented below. The results are expressed as mean fluorescence intensity (MFI) due to CFSE staining. Higher values of fluorescence intensity mean less cell proliferation.

Analyzing the obtained results, it was possible to conclude that the two drug combinations promoted a more pronounced arrest on cell division, compared to the individual drugs of each combination. Particularly, with 5-FU + verapamil, MFI values were 162.3 ± 48.9% and with 5-FU+itraconazole these values were 175.4 ± 50.1%, compared with values of 154.9 ± 30.2% (5-FU) and 169.16 ± 49.5% (itraconazole) obtained with the individual drugs with more effect on cell proliferation arrest of each combination, respectively. 5-Fluorouracil combined with itraconazole was the drug combination that led to the highest MFI values.

It is important to note that high SEM values were obtained, since the two experiments resulted in very different values. However, analyzing graph C, it was possible to conclude that both experiments showed the same tendency of results, in which drug combinations are advantageous relative to the isolated drugs.

## 4. Discussion

In this study, the two potential repurposed drugs in the drug screening assay were verapamil and itraconazole, combined with 5-FU, since they were those that showed better responses (increased reduction of cellular viability), when compared to both individual drugs of each combination.

In particular, verapamil is a calcium channel blocker, mainly a l-type calcium channel blocker, inhibiting the transmembrane flux of calcium ions, being utilized clinically to treat cardiac arrhythmias, angina and cardiomyopathies [31,32]. Besides the known role of ion channels in specialized excitable cells, these channels also play critical roles in cancer pathophysiology by several mechanisms, controlling cancer cell proliferation by regulating several key survival signaling pathways and membrane potential [33]. Verapamil is also known to be a first-generation inhibitor of P-glycoprotein, an important protein present in the cell membrane, that pumps a wide variety of substances out of cells [34]. Thus, when combined with chemotherapeutic agents, this drug may help to promote intracellular drug accumulation. This has been demonstrated in non-small cell lung cancer, colorectal carcinoma, leukemia and neuroblastoma cell lines [30]. Particularly in breast cancer, several studies reported beneficial effects of verapamil. For example, this drug shown anti-proliferative effect in breast cancer in a mouse model [35]. Potentiation of tamoxifen activity by verapamil, in a human breast cancer cell line (MCF-7), has also been reported [36]. However, there are contradictory reports about the anticancer properties of verapamil. For example, in human epidermoid carcinoma cells, verapamil did not inhibit the growth of cells [33]. Additionally, the current use of calcium-channel blockers, such as verapamil, for 10 or more years was associated with higher risks of ductal and lobular breast cancer. Thus, more research is needed to confirm these findings and to evaluate potential underlying biological mechanisms of verapamil in cancer [37].

Verapamil is an l-type calcium channel blocker, but it is also a blocker of other calcium channels. Concerning itraconazole, it is a broad-spectrum anti-fungal agent that inhibits lanosterol 14-α-demethylase, being used to treat fungal infections and for prophylaxis in immunosuppressive disorders [29,38]. Studies have shown that this drug possesses antineoplastic activity and also has a synergistic action when combined with chemotherapeutic agents. Although it showed promising anticancer activity in several types of cancer, its precise anticancer mechanism has remained elusive [29,39]. However, it is known that it acts via several mechanisms to prevent tumor growth, including inhibition of the Hedgehog pathway, prevention of angiogenesis, decreased endothelial cell proliferation, and cell cycle arrest. Also, in ovarian and breast cancer, in vitro studies confirm that itraconazole inhibits P-glycoprotein, thus reversing resistance conferred by this efflux pump [29]. In breast cancer, a pilot trial evaluated the pharmacokinetics of this drug when administered to 13 patients with metastatic breast cancer. The conclusions were that as the plasma levels of itraconazole increased, higher levels of thrombospondin-1, which inhibits angiogenesis, were detected [9]. In another study, itraconazole inhibited MCF-7 and SKBR-3 cells proliferation via induction of cell death and G0/G1 cell cycle arrest [40].

Despite these drugs showing at least minimal evidence of some cancer activity, their mechanisms of action for this condition are not yet fully defined. It is also important to note that in this research, because drugs have been studied in the context of combination with 5-FU, the mechanism by which they act in the cell may be different from that which would be observed with the individual drugs, since the combination may act together in a different way in each of its individual components. Thus, a deep study of these mechanisms may represent an important step in breast cancer therapy, as well as in other types of cancer.

The data obtained in this work allowed to conclude that the two selected drug combinations have potential beneficial effects on breast cancer therapy. Thus, analyzing the obtained results along this research, it was clear that cellular viability was reduced in comparison with individual drugs of the combination, in the great majority of cases. This reduced cellular viability observed by MTT assay, as well as the cellular damage observed in cells treated with drug combinations, namely presence of damage in cell membranes and cellular vacuoles in the cytoplasm (Figure 11) may reflect events such as cell death or cell proliferation arrest.

The preliminary results obtained by CFSE staining of cells indicated that all drug combinations showed a tendency to stop cell proliferation. Thus, it can be hypothesized that the drug combinations may act primarily in the arrest of cell proliferation. The combination 5-FU + itraconazole being the combination of drugs that most affects this parameter. Besides the data obtained by CFSE staining of cells, the data obtained by the comparison of MCF-7 and MCF-10A cell lines also supported this hypothesis; once the drugs had a much more pronounced effect on MCF-7 than MCF-10A cells, the last, the proliferation decrease and, thus, divided less. Additionally, when comparing the results obtained for 48 and 72 h-treatments, the cell viability was more reduced at 72 h, sustaining again the hypothesis that the drug combinations act on cell proliferation; in 72 h, they have more time to proliferate (and, maybe, incorporate the drug), rather than in 48 h. 5-Fluorouracil acts mainly on the S phase of cell cycle, thus, an explanation about this tendency of drug combinations to stop proliferation may be, for example, a potentiation of the 5-FU action by the repurposed drugs or a promotion of the accumulation of 5-FU inside the cells. Further studies to test this hypothesis will be relevant.

Even though not very conclusive (possible mainly because of the small number of experiments), annexin V/PI staining of cells showed that apoptosis was not observed at 3 h of treatment, except in the case of 5-FU + itraconazole (126.8 ± 23.9%), but was observed in the majority of cases (single drugs and combinations) at 8 h, suggesting that with later times, the observation of apoptotic events is clearer. The same happened concerning late apoptosis/necrosis, where an increase in these values was observed at 8 h of treatment, comparing to 3 h. Also, it was expected that in the conditions where apoptosis was verified at 3 h (only verified with 5-FU + itraconazole), the 8 h trend would be an increase in late apoptosis, which was not verified, probably because this process took even longer than 8 h to occur. The timeline of apoptotic-related biochemical events depends on a huge number of factors, such as cell culture conditions, the cell line used, drug concentration/stimulus intensity and exposure time [41]. In cell culture, apoptotic signs may be accomplished in less than two hours [42]. However, there are also studies that report that, in a general way, cultured cells induced to undergo apoptosis exhibit signs of apoptotic events within 5 to 10 h [41]. Combining all results obtained by this technique, it was notorious that there was a greater tendency to cell death at 8 h than at 3 h, either by apoptosis or by necrosis. However, in this moment, it is not possible to conclude that a certain combination acts via apoptosis and/or necrosis, being crucial to increase the number of independent experiments to obtain more sustainable results that can allow better conclusions. Also, concerning this assay, it would be useful to test at different time points, namely later times.

On crossing data from proliferation and cell death assays, the seemingly more plausible conclusion is that drug combinations act essentially on pathways related to cell proliferation arrest. After this arrest, the cells tend to accumulate damage (Figure 11) and, consequently die. Hence, it is plausible that the highest cell death values were obtained for 8 h. This increase is likely to continue with increasing drug exposure time.

Another study carried out in the scope of this work was the evaluation of the EMT of possible drug resistant cells. Epithelial-mesenchymal transition is an important process in normal development but also plays a critical role during tumor progression and malignant transformation, leading to the acquisition of invasive and metastatic properties in cancer cells [43].

Studies have reported that the acquisition of 5-FU resistance induces epithelial-mesenchymal transitions through the Hedgehog signaling pathway in HCT-8 colon cancer cells [44]. Also, EMT was associated with acquired resistance to 5-FU in HT-29 colon cancer cells [45]. Additional studies also proved that chemoresistance to 5-FU induces epithelial-mesenchymal transition via upregulation of Snail in MCF-7 human breast cancer cells [46]. Thus, to study this transition of state, the presence of E-cadherin, cytokeratins AE1/AE3 and vimentin was studied.

The downregulation of E-cadherin is a molecular signature of EMT. This molecule is an adhesion molecule that is present in the plasma membrane of most epithelial cells [47]. Cytokeratins are also markers of an epithelial phenotype. This is a protein with an intermediate filament made of keratin, having important roles in cell differentiation [48,49]. Particularly, cytokeratin AE1/AE3 is a mixture of two different clones of anti-cytokeratin monoclonal antibodies, AE1 and AE3, that detect high and low molecular weight keratins. By combining these two reagents, a single reagent with a broad spectrum of reactivity against a huge diversity of cytokeratins is obtained [49]. Relative to vimentin, it is ubiquitously expressed in normal mesenchymal cells, being known to maintain cellular integrity and providing resistance against stress. Overexpression of vimentin in cancer correlates well with accelerated tumor growth, invasion, and poor prognosis [50].

E-cadherin’s downregulation is an important leading event for EMT and is considered a hallmark of this transition. Its downregulation can lead to decreased expression and/or organization of additional epithelial markers. Concomitantly, increased expression of mesenchymal proteins (such as vimentin), as well as of extracellular matrix remodeling enzymes occurs together with actin cytoskeleton reorganization [51]. Thus, this seems to be a process in which several events occur concomitantly.

Analyzing the images obtained by immunocytochemistry, it was clear that no EMT was observed, in any of the cases. However, in the case of 5-FU combined with verapamil, there seemed to be a very weak staining of E-cadherin. The reason for this finding is unknown, since there are no studies with this drug combination regarding EMT in cancer cells, particularly in breast cancer cells. Thus, this combination may have some action on E-cadherin or even produce a transition from epithelial to mesenchymal state, first observed by a loss of E-cadherin; this may not be fully observed due to a short time of drug action. Thus, it could be stated that 48 h was not enough time for the transition to the mesenchymal state to occur. However, studies show that this transition is observed earlier. For example, in a study using the MCF-7 cell line to study whether EGF induced EMT, it was found that the loss of E-cadherin, as well as increased expression of vimentin, began to be observed about 4–8 h after the stimulus [52]. In another study, EMT occurred on day seven after the stimulus, in this case with granulocyte macrophage colony-stimulating factor for colon cancer [53]. Thus, it is noted that the observation of EMT depends on the stimulus, as well as on the cell lines in which the study is taking place. An interesting future approach could be the study of this transition at different times points.

Before concluding, it is also important to highlight that one of the factors that should be considered in this research work is the concentration of the drugs used in most of the experiments, which was based on the concentration-effect curves and, more specifically, on the IC_50_ values obtained. The IC_50_ concept is not clear. For example, it would be ambiguous if the values that define 100% and 0% are not clearly defined [54,55]. To solve this problem, normalizing the data in order that responses vary between 0 and 100, forcing the bottom and top plateaus to equal 0 and 100, respectively, may be a solution. However, a normalized model should be applied only when the values that define 0 and 100 are correctly determined [54,55]. In this project, IC_50_ values of the three repurposed drugs was calculated considering this type of normalization, since no perfectly defined dose-response curves with defined plateaus and where both maximal and minimal activity were very explicit. Nevertheless, the obtained value for the IC_50_ of the reference drug, 5-FU was consistent with that obtained in studies of this drug in MCF-7 using the same methodology (MTT) [56]. Thus, concerning obtained IC_50_ values, itraconazole appeared to be the most potent drug, since it presented the lowest IC_50_ value.

Another important aspect is the fact that only one ratio of concentrations (1:1) between the combinations was tested in the screening assay (Figure 1), because it was important to ensure that the mechanisms of action of both drugs did not overlap. In line with this, for example, in vitro experiments identified a strong antagonism between irinotecan and cisplatin when they were administered at a 1:1 ratio, but in a ratio of 4:1, a synergistic effect was observed. Also, the combination of irinotecan and floxuridine was synergistic at an equimolar ratio but was strongly antagonistic at a 10:1 ratio [57]. Thus, it is a hard process to optimize ratios of concentration in drug combinations.

Finally, by integrating all the obtained results, it was noted that all drug combinations showed a tendency to be more effective than the most effective drug of each combination. The best results were obtained for 72-h treatment and, in most cases, there were no accented differences between the different concentrations tested.

Although the combination of 5-FU and verapamil was the one that led to the lowest cell viability values (approximately 12%, at 72 h in concentrations of 50 μM of each drug in the combination), the collective of all results pointed out that the combination of 5-FU and itraconazole was the most promising combination. The results obtained by MTT reduction assay support this evidence (both at 48 h and at 72 h, regardless of the concentration used, except at 72 h, at concentrations of 50 μM of each drug in the combination, in which case 5-FU + verapamil was highlighted). Furthermore, analyzing the IC_50_ values obtained, itraconazole was the drug that presented the lowest value, being apparently the most potent. Additionally, CFSE staining of cells showed that 5-FU combined with itraconazole was the drug combination that promoted more elevated values of MFI, indicating a more pronounced proliferation arrest. Studies in MCF-10A cells also supported this hypothesis, since itraconazole was the only drug which when combined with 5-FU and used individually showed significantly different effects relative to the control, which may indicate increased toxicity. In addition to all this, it is important to mention that concerning the two potential repurposed drugs used in this research, itraconazole is the one that is most studied in the field of cancer, showing promising results. Additionally, this drug was used in a lower concentration and, as known, a successful therapy is a therapy in which efficacy is achieved with the lowest possible drug concentration. Thus, the two drug combinations were shown to be promising in breast cancer therapy, even though evidence pointed to itraconazole, and particularly its combination with 5-FU, a very relevant object of study for the therapy of breast cancer and, possibly, other types of cancer.

## 5. Concluding Remarks

Cancer is a complex group of diseases that involves several pathways and different molecules, being very challenging to treat. Oncological therapy, which includes breast cancer therapy, is increasingly being studied with therapeutic strategies such as drug repurposing and drug combination being largely adopted.

In this research, the combination of drugs with potential to be repurposed (verapamil, itraconazole) with a reference chemotherapeutic drug (5-FU) showed advantages over individual drugs, mainly by arresting cell proliferation and decreasing cellular viability. Thus, the drug combinations showed evidence to be promising for the treatment of breast cancer. A future study on more resistant breast cancer cell lines (such as the triple negative line MDA-MB-231) and also on different cancer cell lines corresponding to other types of cancer would be interesting. In vivo studies for application in human therapy would also be extremely important in a more advanced line of work.

Based on recent literature [58,59,60,61], we believe that optimizing drug ratios could result in better antitumor efficacy than the single-drug system and each drug significantly changes synergy between the drugs. However, for all of this to be achieved, a better knowledge about the mechanisms of action underlying these drug combinations is required, with a special focus on 5-FU combined with itraconazole, since it was, globally, the most promising drug combination.

## Figures and Tables

**Figure 1 biomolecules-08-00175-f001:**
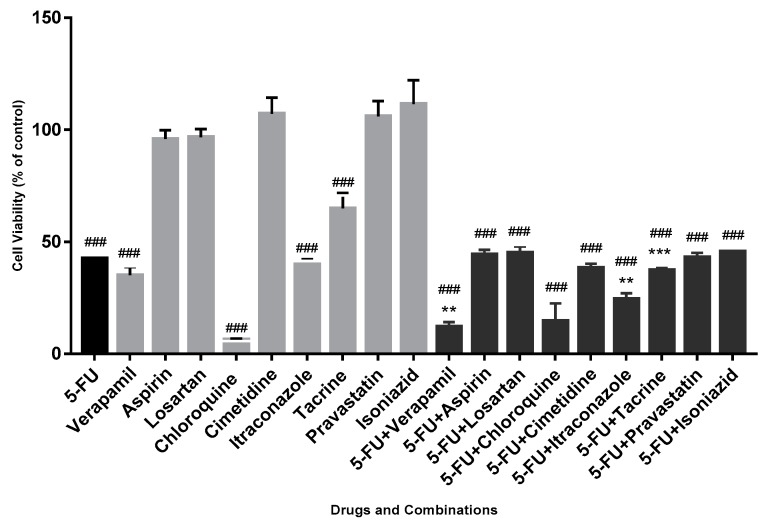
Effect of drugs and respective drug combinations on the viability of MCF-7 cells. Each drug and each combination were added in fresh medium, in sextaplicates. Results are presented as mean ± standard error of the mean (SEM) and represent the viability of cells (% of control) of 3–4 independent experiments (*n* = 3, 4). ### *p* < 0.001 vs. control; ** *p* < 0.01 and *** *p* < 0.001 vs. single drug of the combination with more effect on cell viability reduction. 5-FU: 5-fluorouracil.

**Figure 2 biomolecules-08-00175-f002:**
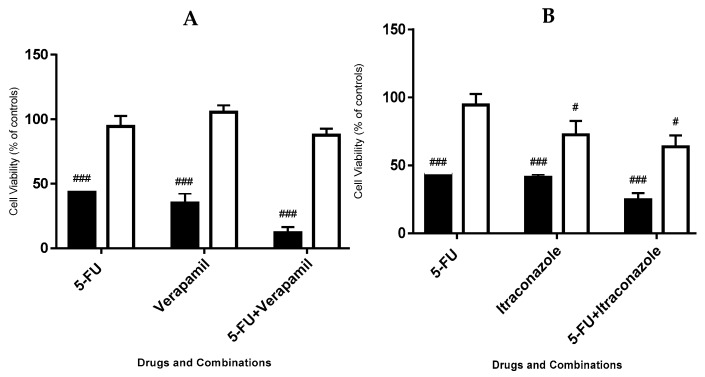
Effect of 5-FU, verapamil (**A**), itraconazole (**B**) and respective combinations with 5-FU on the viability of MCF-7 cells (left, black bars) and MCF-10A cells (right, white bars). Each drug and combination were added in fresh medium, in sextaplicates. Results are presented as mean ± SEM and represent the viability of cells (% of control) of 3–4 independent experiments (*n* = 3, 4). ### *p* < 0.001 and # *p* < 0.05 vs. respective controls.

**Figure 3 biomolecules-08-00175-f003:**
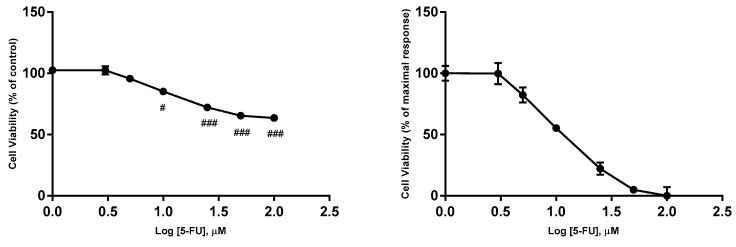
Concentration-effect curves obtained for 5-FU. The drug was added in fresh medium, in sextaplicates. Results are presented as mean ± SEM of three independent experiments. The left curves represent the viability of cells (% of control), whereas the right curves represents a normalization between 0 and 1 (0% and 100%), where 100% was defined as the concentration of the drug that less affected the cell viability, and all other data points were normalized to this value, being 0% defined as the concentration that affected more the cell viability. ### *p* < 0.001 and # *p* < 0.05 vs. control.

**Figure 4 biomolecules-08-00175-f004:**
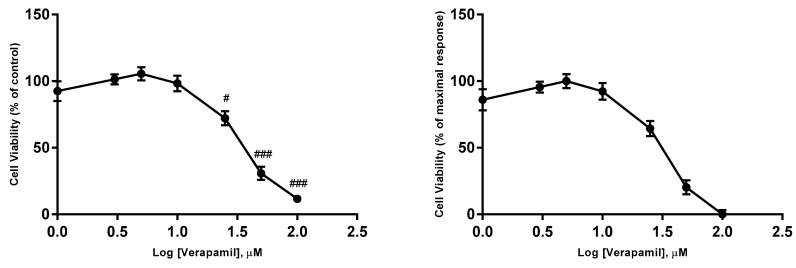
Concentration-effect curves obtained for verapamil. The drug was added in fresh medium, in sextaplicates. Results are presented as mean ± SEM of three independent experiments. The left curve represents the viability of cells (% of control), whereas the right curve represents a normalization between 0 and 1 (0% and 100%), where 100% was defined as the concentration of the drug that less affected the cell viability, and all other data points were normalized to this value, being 0% defined as the concentration that affected more the cell viability. ### *p* < 0.001 and # *p* < 0.05 vs. control.

**Figure 5 biomolecules-08-00175-f005:**
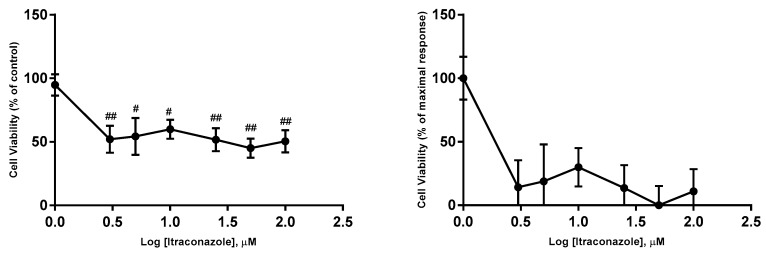
Concentration-effect curves obtained for itraconazole. The drug was added in fresh medium, in sextaplicates. Results are presented as mean ± SEM of five independent experiments. The left curve represents the viability of cells (% of control), whereas the right curve represents a normalization between 0 and 1 (0% and 100%), where 100% was defined as the concentration of the drug that less affected the cell viability, and all other data points were normalized to this value, being 0% defined as the concentration that affected the cell viability more. ## *p* < 0.01 and # *p* < 0.05 vs. control.

**Figure 6 biomolecules-08-00175-f006:**
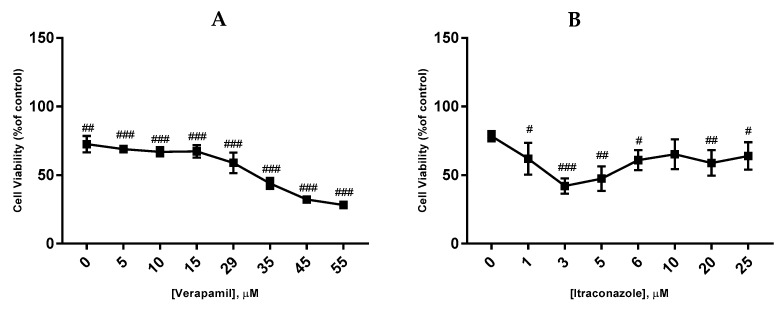
Effect of verapamil (**A**) and itraconazole (**B**) combined with 5-FU on the viability of MCF-7 cells. Each drug and drug combination were added in fresh medium, in sextaplicates. Results are presented as mean ± SEM, and represent the viability of cells (% of control) of three independent experiments. ### *p* < 0.001, ## *p* < 0.01 and # *p* < 0.05 vs. control.

**Figure 7 biomolecules-08-00175-f007:**
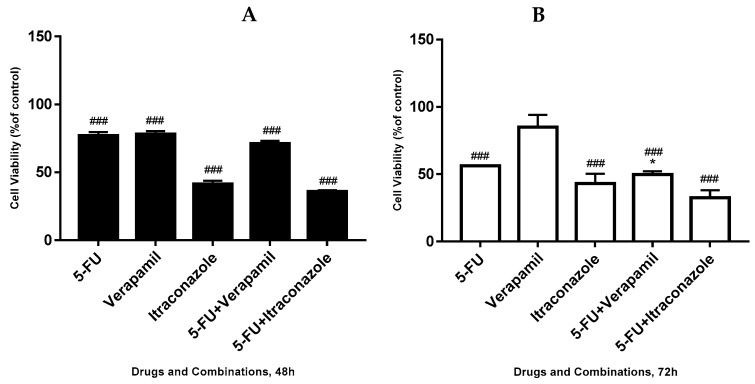
Effect of 5-FU, verapamil, itraconazole, and respective combinations of the last two drugs with 5-FU on the viability of MCF-7 cells, for 48 h (**A**), 72 h (**B**). Each drug and drug combination were added in fresh medium, in sextaplicates. Results are presented as mean ± SEM, and represent the viability of cells (% of control) of four independent experiments. ### *p* < 0.001 vs. control; * *p* < 0.05 vs. single drug of the combination with more effect on cell viability reduction.

**Figure 8 biomolecules-08-00175-f008:**
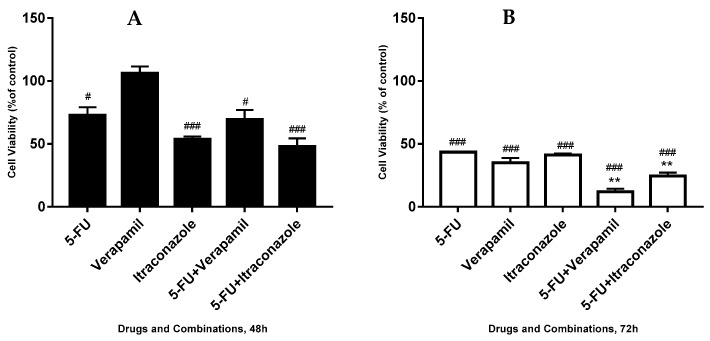
Effect of 5-FU, verapamil, itraconazole, and respective combinations of the last two drugs with 5-FU on the viability of MCF-7 cells, for 48 h (**A**), 72 h (**B**) and a comparison of both times (**C**). Each drug and drug combination were added in fresh medium, in sextaplicates. Results are presented as mean ± SEM, and represent the viability of cells (% of control) of 3–4 independent experiments. ### *p* < 0.001 and # *p* < 0.05 vs. control; ** *p* < 0.01 vs. single drug of the combination with more effect on cell viability reduction.

**Figure 9 biomolecules-08-00175-f009:**
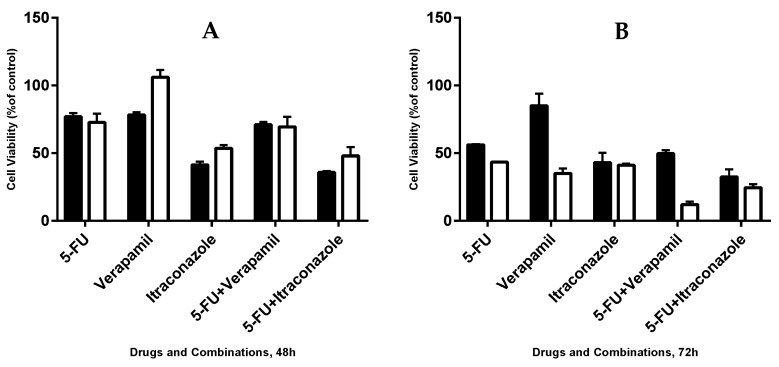
Effect of 5-FU, verapamil, itraconazole, and respective combinations of the last two drugs with 5-FU on the viability of MCF-7 cells, for 48 h (**A**) and 72 h (**B**). In the left, black bars, 5-FU was used in a concentration of 11.8 μM and verapamil and itraconazole were used in a concentration of 55 and 3 μM, respectively. In the right, white bars, all drugs were used in a concentration of 50 μM. Each drug and drug combination were added in fresh medium, in sextaplicates. Results are presented as mean ± SEM, and represent the viability of cells (% of control) of 3–4 experiments.

**Figure 10 biomolecules-08-00175-f010:**
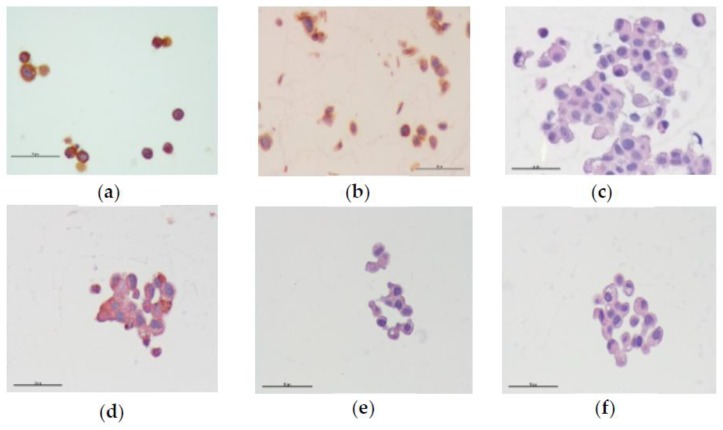
Expression of EMT markers, AE1/AE3, E-cadherin and vimentin in MCF-7 cells, by immunocytochemistry. All images were obtained through the same pool of cells, but in different cell blocks, in a magnification of 400x. (**a**–**c**) represent the positive controls for AE1/AE3, E-cadherin and vimentin, respectively. (**d**–**f**) represent images of cells treated with 5-FU combined with verapamil and tested for the presence of AE1/AE3, E-cadherin and vimentin, respectively. Images of cells treated with 5-FU combined with itraconazole and tested for the presence of AE1/AE3, E-cadherin and vimentin are represented by (**g**–**i**), respectively. Scale bar: 50 μm.

**Figure 11 biomolecules-08-00175-f011:**
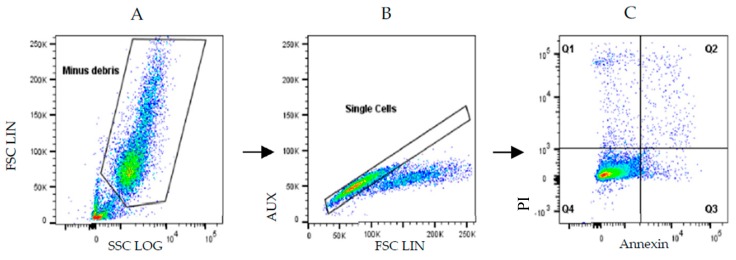
Flow cytometry gating strategy used to evaluate apoptosis/necrosis in MCF-7 cells. (**A**) The gate separated MCF-7 cells from cellular debris, whereas in (**B**) single cells are gated, thus excluding aggregates. (**C**) Target cells were distinguished based on PI/Annexin V staining. In Q1 + Q2 are represented cells in necrosis/late apoptosis, in Q3 are represented cells that are in early apoptosis and living cells are represented in Q4. SCC: side-scattered light; FSC: forward-scattered light; AUX: auxiliar; LIN: linear scale; Log: logarithimic scale; PI: propidium iodide

**Figure 12 biomolecules-08-00175-f012:**
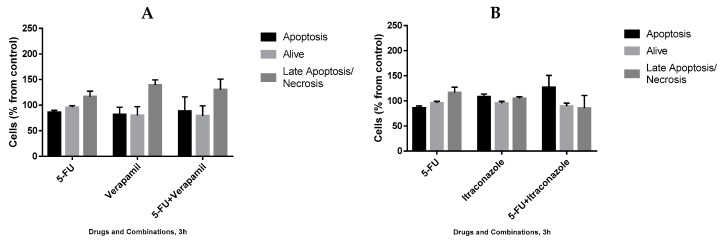
Effect of 5-FU, verapamil (**A**), itraconazole (**B**), and respective combinations of the last two drugs with 5-FU on the death of MCF-7 cells, for 3 h. Each drug and drug combination were added in fresh medium, in duplicates. Results are presented as mean ± SEM of two independent experiments.

**Figure 13 biomolecules-08-00175-f013:**
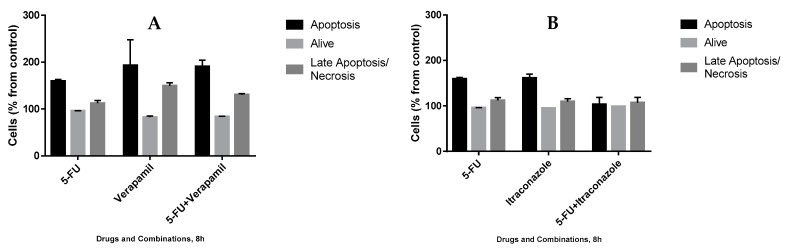
Effect of 5-FU, verapamil (**A**), itraconazole (**B**), and respective combinations of the last two drugs with 5-FU on the death of MCF-7 cells, for 8 h. Each drug and drug combination were added in fresh medium, in duplicates. Results are presented as mean ± standard deviation (SD) of a single experiment.

**Figure 14 biomolecules-08-00175-f014:**
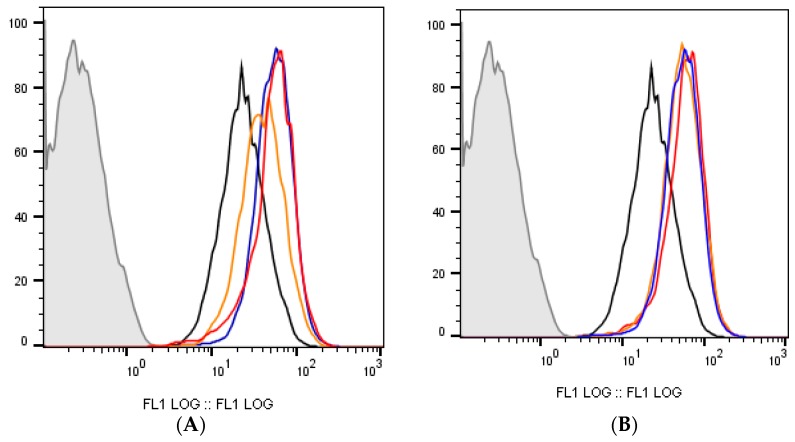
Cell fluorescence due to CFSE staining. In all graphs, the black line indicates control (stained), the grey line indicates unstained control and the blue line represents 5-FU. (**A**) The red line indicates the combination of 5-FU + verapamil and the orange line indicates verapamil. (**B**) Red and orange lines represent 5-FU + itraconazole and itraconazole, respectively.

**Figure 15 biomolecules-08-00175-f015:**
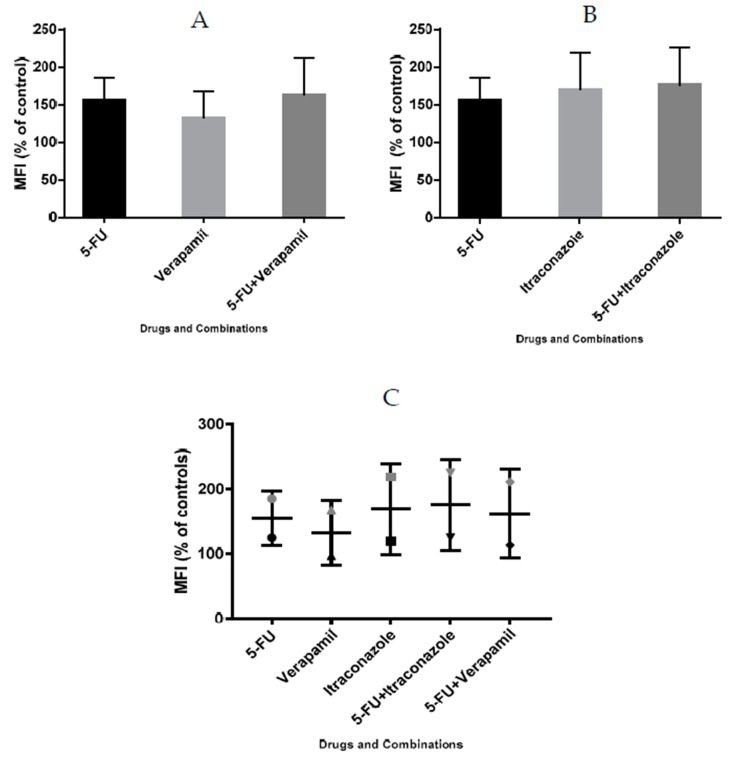
Effect of 5-FU, verapamil (**A**), itraconazole (**B**), and respective combinations of the last two drugs with 5-FU on the proliferation of MCF-7 cells, and a comparison of both experiments (**C**). Each drug and drug combination was added in fresh medium, in duplicates. Results are presented as mean ± SEM, and represent median fluorescence intensity (MFI) (% of control) of 2 independent experiments.

**Figure 16 biomolecules-08-00175-f016:**
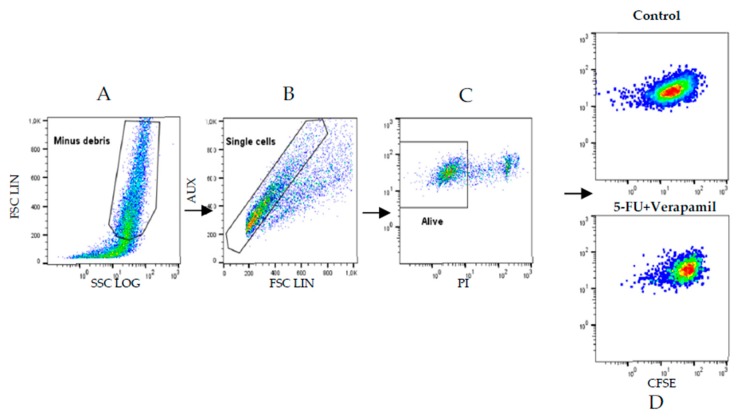
Flow cytometry gating strategy used in the carboxyfluorescein succinimydil ester (CFSE) proliferation assay of MCF-7 cells. (**A**) The gate separated MCF-7 cells from cellular debris, whereas in (**B**) single cells are gated, thus excluding aggregates. (**C**) Dead cells were excluded by PI incorporation, and (**D**) represents cell fluorescence due to CFSE staining in non-treated (control) or 5-FU + Verapamil (treated cells), as indicated.

**Table 1 biomolecules-08-00175-t001:** Obtained IC_50_ values for 5-FU, verapamil and itraconazole on the viability of MCF-7 cells, with the corresponding 95% confidence intervals.

Drug	IC_50_ (μM)
5-FU	11.79
Verapamil	29.49
Itraconazole	2.08

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
