# Peer review of "Study of New Therapeutic Strategies to Combat Breast Cancer Using Drug Combinations"

_biomolecules, 2018, doi:10.3390/biom8040175_

Round 1
Reviewer 1 Report
The Manuscript entitled “Study of New Therapeutic Strategies to Combat Breast Cancer using Drug Combinations” written by Correia et. al demonstrates the combination of drugs with potential to be repurposed (verapamil, itraconazole) with a reference chemotherapeutic drug (5-FU) showed advantages over the individual drugs, mainly by arresting cell proliferation and decreasing cellular viability. This article is certainly worth publishing with minor corrections. The following remarks are suggested for consideration of the authors when they prepare the final version of the article.
Authors should include graphical abstract as well.
Authors should discuss why 5-FU was chosen as the reference drug of this study in “Introduction”.
The authors should state clearly what is the takeaway message from the cell viability study like why the combination of 5-FU and verapamil was the one that led to the lowest cell viability values whereas the collective of all results pointed out that the combination of 5-FU and itraconazole was the most promising combination.
Author Response
The Manuscript entitled “Study of New Therapeutic Strategies to Combat Breast Cancer using Drug Combinations” written by Correia et. Al demonstrates the combination of drugs with potential to be repurposed (verapamil, itraconazole) with a reference chemotherapeutic drug (5-FU) showed advantages over the individual drugs, mainly by arresting cell proliferation and decreasing cellular viability. This article is certainly worth publishing with minor corrections. The following remarks are suggested for consideration of the authors when they prepare the final version of the article.
We thank the reviewer for their frankly positive opinion on our work, as well as for their careful revision, comments, and suggestions, which helped us to significantly improve our manuscript. The latter has been revised to meet the reviewers’ comments and suggestions, as follows.
- Authors should include graphical abstract as well.
Yes, we agree and the graphical abstract was introduced.
- Authors should discuss why 5-FU was chosen as the reference drug of this study in “Introduction”.
We have amended the sentence, and thank the Reviewer for the correction. Now, we can read on page 3, the following text:
“5-FU was chosen as the reference drug of this study, mainly because it is a potential drug to be used in combination regimens in breast cancer therapeutics, with the major aim of improving its efficacy, as well as its known toxicological profile. It is a heterocyclic aromatic organic compound with a structure similar to that of the pyrimidine molecules of DNA and RNA. This drug and their metabolites are responsible for RNA and DNA damage, acting on S phase of cell cycle [27].
Studies addressing combinations of drugs using drugs with non-cancer indications have shown very positive results in relation to cancer therapy, not exclusively for breast cancer, but also for other kinds of cancer [28].”
[27] Longley, D.B., D.P. Harkin, and P.G. Johnston, 5-fluorouracil: mechanisms of action and clinical strategies. Nature Reviews Cancer, 2003. 3(5): p. 330. DOI: 10.1038/nrc1074
[28] Hu, Q., et al., Itraconazole induces apoptosis and cell cycle arrest via inhibiting Hedgehog signaling in gastric cancer cells. Journal of Experimental & Clinical Cancer Research, 2017. 36: p. 50. DOI: 10.1186/s13046-017-0526-0
- The authors should state clearly what is the takeaway message from the cell viability study like why the combination of 5-FU and verapamil was the one that led to the lowest cell viability values whereas the collective of all results pointed out that the combination of 5-FU and itraconazole was the most promising combination.
This observation is correct. However, there is greater consistency in the results with itraconazole in several experiments. It was with this drug that a lower IC50 of all, including 5-FU, was obtained.
Reviewer 2 Report
In this work, Correia et al. reported the screening of drug combinations for breast cancer treatment. The topic of this work is worthy of investigation. However, the presentation of the results in this manuscript is very messy and hard to follow. Especially, there is a huge issue of the manuscript: The same data has been presented multiple times in multiple graphs. The manuscript requires significant revision according to the following comments.
Major concerns:
1. Same data should be only presented only once in the manuscript. There are too many duplicated data in the manuscript. For example, same data was presented in Fig2&3, Fig8A,B&C, Fig9A,B&C et al. The authors should make sure all the duplicated data being removed.
2. Drug synergy has been reported to be largely dependent on the ratio of drugs used. However, in this study, especially in Figure 1, the authors only tested one drug ratio and made conclusions based on this. This is actually not right. The authors should clarify why they only used one ratio. Also, the authors should also briefly discuss the ratio-dependent drug synergy and cite the following relevant literature, such as Bioengineering & Translational Medicine 2018, 3(1): 49-57; Journal of Controlled Release 2017, 267: 191-202; Molecular Pharmaceutics 2017, 14(8): 2697-2710; ACS Omega 2018, 3(8): 9210-9219 et al.
Author Response
In this work, Correia et al. reported the screening of drug combinations for breast cancer treatment. The topic of this work is worthy of investigation. However, the presentation of the results in this manuscript is very messy and hard to follow. Especially, there is a huge issue of the manuscript: The same data has been presented multiple times in multiple graphs. The manuscript requires significant revision according to the following comments.
We thank the reviewer for the observations; we also acknowledge the time spent in the review and the perceptive comments that will undoubtedly improve the quality of the manuscript. The response is addressed below.
Major concerns:
1. Same data should be only presented only once in the manuscript. There are too many duplicated data in the manuscript. For example, same data was presented in Fig2&3, Fig8A,B&C, Fig9A,B&C et al. The authors should make sure all the duplicated data being removed.
Yes, we agree. We thank the reviewer for the observations. Indeed, a new versions for Figures 8 and 9 were prepared. Also, Figure 2 was removed from the manuscript.
2. Drug synergy has been reported to be largely dependent on the ratio of drugs used. However, in this study, especially in Figure 1, the authors only tested one drug ratio and made conclusions based on this. This is actually not right. The authors should clarify why they only used one ratio.
Yes, we understand. In Page 19 we can read now:
“Another important aspect is the fact that only one ratio of concentrations (1:1) between the combinations was tested in the screening assay (Figure 1), because it was important to ensure that the mechanisms of action of both drugs did not overlap to each other. In line with this fact, for example, in vitro experiments identified a strong antagonism between irinotecan and cisplatin when they were administered at a 1:1 ratio, but in a ratio of 4:1, synergistic effect was observed. Also, the combination of irinotecan and floxuridine was synergistic at an equimolar ratio but was strongly antagonistic at a 10:1 ratio [60]. Thus, it is a hard process to optimize ratios of concentration in drug combinations.”
[60] Tolcher, A.W. and L.D. Mayer, Improving combination cancer therapy: the CombiPlex((R)) development platform. Future Oncology, 2018. 14(13): p. 1317-1332.
Also, the authors should also briefly discuss the ratio-dependent drug synergy and cite the following relevant literature, such as Bioengineering & Translational Medicine 2018, 3(1): 49-57; Journal of Controlled Release 2017, 267: 191-202; Molecular Pharmaceutics 2017, 14(8): 2697-2710; ACS Omega 2018,3(8): 9210-9219 et al.
We thank the reviewer for this topic (drug synergism). Indeed, a new text was introduced in the manuscript (page 20):
“Based on recent literature [61-64], we believe that optimizing drug ratios could result in better antitumor efficacy than the single-drug system and each drug significantly changes synergy between the drugs”.
[61] Vogus, D.R., et al., Schedule dependent synergy of gemcitabine and doxorubicin: Improvement of in vitro efficacy and lack of in vitro-in vivo correlation. Bioengineering & Translational Medicine, 2018. 3(1): p. 49-57. DOI: 10.1002/btm2.10082.
[62] Vogus, D.R., et al., A hyaluronic acid conjugate engineered to synergistically and sequentially deliver gemcitabine and doxorubicin to treat triple negative breast cancer. Journal of Controlled Release, 2017. 267: p. 191-202. DOI: 10.1016/j.jconrel.2017.08.016.
[63] Zhao, Z., et al., A nano-in-nano polymer-dendrimer nanoparticle-based nanosystem for controlled multidrug delivery. Molecular Pharmaceutics, 2017. 14(8): p. 2697-2710. DOI: 10.1021/acs.molpharmaceut.7b00219.
[64] Lou, S., et al., Multifunctional nanosystem for targeted and controlled delivery of multiple chemotherapeutic agents for the treatment of drug-resistant breast cancer. ACS Omega. 2018. 3(8): p. 9210-9219. DOI: 10.1021/acsomega.8b00949.
Round 2
Reviewer 2 Report
Ready to publish.